# Biohydrogen production beyond the Thauer limit by precision design of artificial microbial consortia

İpek Ergal[1], Oliver Gräf[1], Benedikt Hasibar [2], Michael Steiner[1], Sonja Vukotić[1], Günther Bochmann [2], Werner Fuchs[2] & Simon K.-M. R. Rittmann [1✉]

Dark fermentative biohydrogen ($H_2$) production could become a key technology for providing renewable energy. Until now, the $H_2$ yield is restricted to 4 moles of $H_2$ per mole of glucose, referred to as the "Thauer limit". Here we show, that precision design of artificial microbial consortia increased the $H_2$ yield to 5.6 mol mol$^{-1}$ glucose, 40% higher than the Thauer limit. In addition, the volumetric $H_2$ production rates of our defined artificial consortia are superior compared to any mono-, co- or multi-culture system reported to date. We hope this study to be a major leap forward in the engineering of artificial microbial consortia through precision design and provide a breakthrough in energy science, biotechnology and ecology. Constructing artificial consortia with this drawing-board approach could in future increase volumetric production rates and yields of other bioprocesses. Our artificial consortia engineering blueprint might pave the way for the development of a $H_2$ production bioindustry.

[1] Archaea Physiology & Biotechnology Group, Department of Functional and Evolutionary Ecology, Universität Wien, Wien, Austria. [2] Department IFA Tulln, Institute for Environmental Biotechnology, University of Natural Resources and Life Sciences, Wien, Austria. ✉email: simon.rittmann@univie.ac.at

Microorganisms thrive in almost all habitats on Earth, where they fulfil important ecosystem functions as complex and highly dynamic microbial communities[1,2]. Microbial communities exist in high levels of biodiversity, enabling cooperation and interaction among its members in functional metabolic networks[3]. Compared with mono-cultures, a microbial consortium empowers complex metabolic tasks due to the multitude of possible metabolic reactions and interaction possibilities, which are based on mutualism, commensalism or neutralism[4,5]. The streamlined syntrophic interactions or commensal relationships among the microorganisms in microbial consortia were shown to enable an efficient utilization of unrefined substrates, such as cane molasses or beet molasses[6,7], to resist to environmental stressors, e.g., temperature fluctuations or heavy metal exposure[7–9], and to display high productivity or yield[10,11]. In nature, a modest undefined consortium may contain thousands of species[12]. However, for efficiently performing bioconversions in natural or artificial ecosystems, the specific metabolic reactions of individual species in the consortium are more relevant than the species richness[13,14].

In environmental, biopharmaceutical or energy biotechnology, most of the bioprocesses are developed and optimized through targeted bioprocess development, utilizing metabolically engineered or wild-type organisms, or even undefined microbial consortia of organisms. The emphasis lies in the optimization of productivity and/or yield by using different types of bioreactors and organisms/undefined consortia. However, every organism, even a metabolically engineered organism, possesses specific metabolic bottlenecks, which limit a full substrate to target product conversion. In many cases, the production of the target compound is accompanied by excretion of several metabolic byproducts, which balance cellular homoeostasis, reducing yield and/or productivity. Moreover, bioprocess development relies on established bioreactors and cultivation pipelines.

Synthetic or artificial microbial consortia are regarded as part of the solution to debottleneck the inherent physiological limitations of wild-type or metabolically engineered mono-culture and undefined consortia bioprocesses, such as enabling the breakdown of complex carbon sources[15], efficient substrate utilization[16], reducing byproduct inhibition through operational stability[17] and high productivities[18]. This can be achieved through selection, design and assembly of microorganisms with specific metabolic (e.g., cellulose utilisers) or ecological (e.g., biofilm forming) functions. In addition, by employing an artificial consortium of selected microorganisms, precision design of a defined microbial co- or multi-culture provides a comprehensive understanding of organismal interactions and allows examining the molecular and eco-physiological basis of community-level functions[19,20]. The developments in the field of artificial microbial ecosystem engineering allowed advancing in the aspects of ecology, such as soil bioremediation[21] and biotechnology, e.g., fine chemical[22,23], biopolymer[24], enzyme[25], food additive[26], antimicrobial[27], biofuel[28] and biohydrogen production[29–31]. However, to achieve supreme efficiency of the bioprocess, a precision design strategy to form an artificial consortium of selected microorganisms was not yet considered.

Molecular hydrogen ($H_2$) is considered as an alternative source of energy. Biological production of $H_2$, referred as biohydrogen production, provides a sustainable and environmentally friendly method for energy generation[32–34]. Dark fermentative $H_2$ production is promising due to high $H_2$ evolution rates (HERs) compared to photobiological $H_2$ production processes[32,33]. However, the yield of $H_2$ per substrate consumed ($Y_{(H2/S)}$) is limited by metabolic constraints of dark fermentative $H_2$-producing microorganisms. According to the theoretical limit, the so-called "Thauer limit", 4 mol $H_2$ can be produced per 1 mol of glucose consumed during dark fermentation when acetate is produced as the byproduct[35]. Depending on the microbial group, $H_2$ formation may occur either via the pyruvate-formate-lyase (PFL) pathway or the pyruvate ferredoxin oxidoreductase (PFOR) pathway[32]. The PFL pathway is operative in Enterobacteriaceae. In this pathway, pyruvate is converted into acetyl-CoA and formate. Formate is either shuttled out of the cell or it can be split into carbon dioxide ($CO_2$) and $H_2$ by formate hydrogen lyase[32]. The PFOR pathway is operative in Clostridiaceae during $H_2$ production, which occurs through the action of [NiFe]- and/or [FeFe]-hydrogenases[36,37]. Up to now, dark fermentative biohydrogen producing wild-type or metabolically engineered mono-cultures were not successful in improving $Y_{(H2/S)}$ beyond the Thauer limit[29,38,39]. Therefore, to boost $Y_{(H2/S)}$, undefined microbial consortia or defined co- and multi-cultures of $H_2$-producing microbes were examined in complex or defined medium[40–43]. However, control of microbial community composition, media compounds and their concentration through precision design of an artificial microbial consortium were not yet the focus of any study.

In our quantitative analysis of pure culture dark fermentative $H_2$ production, we linked physiological and biotechnological characteristics of $H_2$-producing microorganisms through comprehensive meta-data analysis and modelling[32]. Our analysis revealed that Enterobacteriaceae exhibit very high HERs and Clostridiaceae are mesophilic organisms with the highest reported $Y_{(H2/S)}$ on a C-molar level on saccharides. Therefore, we hypothesized that precision design of an artificial microbial consortium composed of Enterobacteriaceae and Clostridiaceae improves $Y_{(H2/S)}$ beyond the Thauer limit.

Here we present results from a drawing board-like precision design of artificial microbial consortium of microorganisms with improved HER, and $Y_{(H2/S)}$ beyond the Thauer limit, of two $H_2$-producing species, the facultative anaerobic *Enterobacter aerogenes* and the obligate anaerobic *Clostridium acetobutylicum*. For the design of this defined artificial consortium, three different major community function-determining parameters were individually and syntrophically investigated: initial substrate concentration of glucose or cellobiose, designing and optimizing a mutual medium, and control of the activity and concentration of initial cell densities. First, initial optimum substrate concentration was investigated for individual strains and a mutual defined medium was designed by applying Design of experiments (DoE). Then, different consortia were created using active inoculum with different initial cell densities of each microorganism. Our interdisciplinary research combines physiology, ecology and biotechnology, provides valuable insights into the ecosystem functionality and enhances $H_2$ production by constructing a defined artificial consortium.

## Results

**Optimizing the initial substrate concentration**. The first step of assembling the artificial consortium was optimizing the initial glucose concentration and to identify the essential nutritional compounds with each of the mono-cultures to prevent substrate inhibition on $H_2$ production[44,45]. To be able to subsequently design the mutual medium, *E. aerogenes* and *C. acetobutylicum* were grown separately in their own microorganism-specific medium and different initial glucose concentrations ranging from 5 to 35 g L$^{-1}$ (Supplementary Fig. 1). Highest $OD_{600}$ of 1.4 and 1.5, and cumulative pressure of 3 and 6.8 bar from *E. aerogenes* and *C. acetobutylicum*, respectively, were measured at a glucose concentration of 30 g L$^{-1}$. Substrate inhibition was observed at the concentrations higher than 30 g L$^{-1}$. Therefore, all of the further experiments were conducted at a concentration of 166.5 mmol L$^{-1}$ (30 g L$^{-1}$

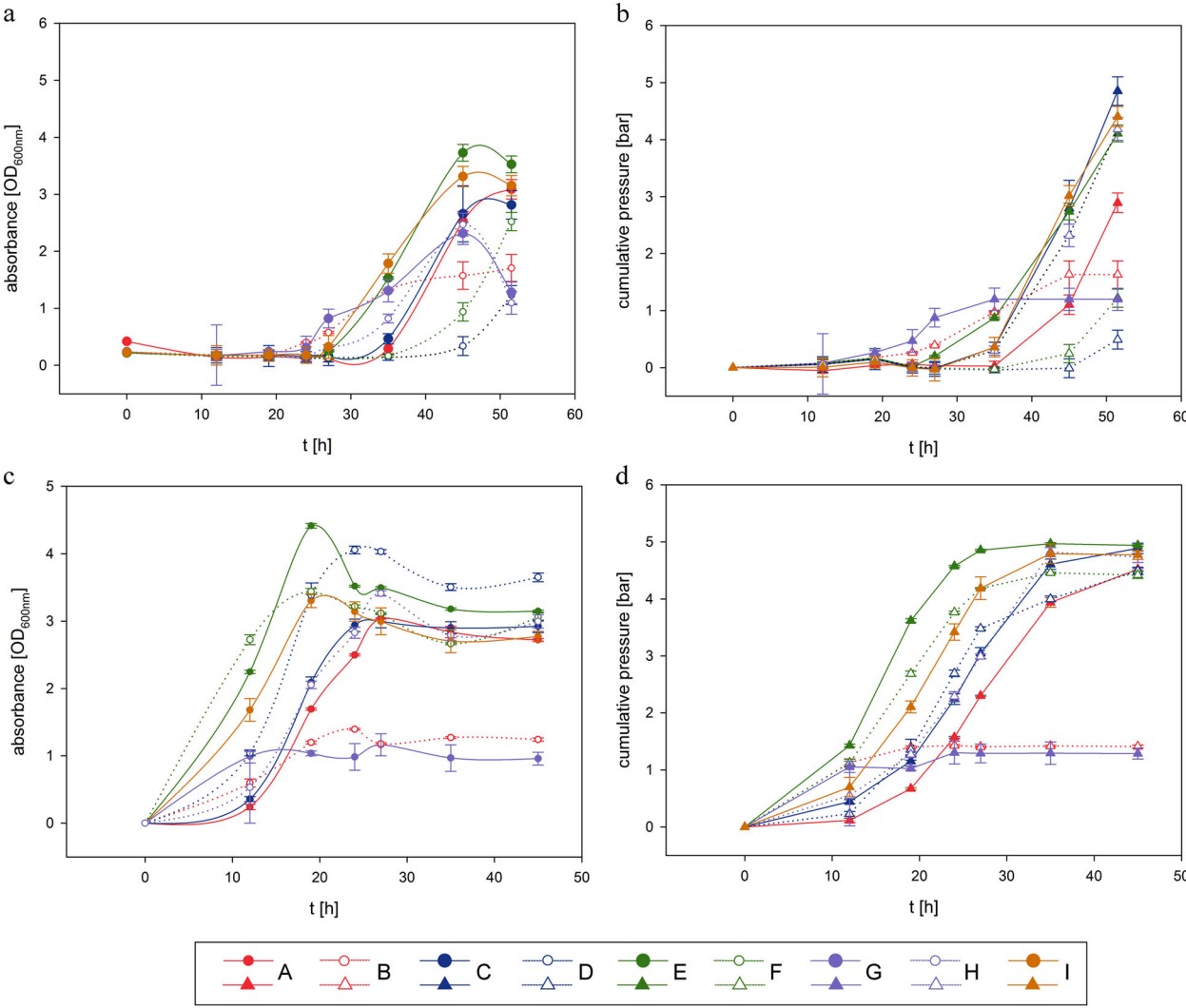

**Fig. 1 Optical density and cumulative pressure measurements of *C. acetobutylicum* and *E. aerogenes* on different DoE media.** Optical density of *C. acetobutylicum* and *E. aerogenes* are shown in **a** and **c**, respectively. Cumulative pressure of *C. acetobutylicum* and *E. aerogenes* are shown in **b** and **d**, respectively.

or 999 carbon-mmol (C-mmol) $L^{-1}$) glucose or at same the carbon level equivalence of cellobiose 83.3 mmol $L^{-1}$ (28.5 g $L^{-1}$ or 999 C-mmol $L^{-1}$).

**Mutual medium design and optimization**. Each microorganism was tested in the medium of the other organisms at a glucose concentration of 30 g $L^{-1}$. Although *E. aerogenes* displayed approximately twofold lower cell density (max. $OD_{600}$ of 0.6) and gas production (cumulative pressure of 1.3 bar) in *Clostridium*-specific medium compared to *Enterobacter*-specific medium, *C. acetobutylicum* did not grow in *Enterobacter*-specific medium (Supplementary Fig. 2). Hence, it was necessary to precision design a mutual medium to accommodate the nutritional needs of both microorganisms, with an emphasis examining phosphate buffer (PB) capacity, ammonium chloride (AC) concentration, and sodium acetate (SA) concentration. These factors were investigated in a DoE setting that compromised eight sets of runs (triplicate) and one set of an additional run (pentaplicate) (A, B, C, D, F, G, H, I and E) (Fig. 1). Concentrations of aforementioned compounds in DoE media are presented in Table 1. The highest $OD_{600}$ values of *C. acetobutylicum* and *E. aerogenes* were observed when E-medium was used (Fig. 1a, c). Even though gas production reached higher values in other medium compositions

(medium; C, H and I), E-medium was superior due to an earlier onset of gas production by *E. aerogenes* (Fig. 1b, d). Through analyses of the specific growth rate ($\mu$) and cumulative gas production, the physiological response of each of the organisms to the multi-parameter settings was modelled. The overlay of the response surface plots, visualizing both models at the same time, is shown in Fig. 2. Models for cumulative pressure and $\mu_{mean}$ for each of the microorganisms indicated that only AC and PB significantly contributed to the model significance. The model for cumulative pressure ($R^2 = 0.89$, $p$-value = <0.0001, Supplementary Table 1) of *C. acetobutylicum* is based on an optimum AC concentration, which is due to a quadratic model term, and on a linear dependence of the PB capacity (Supplementary Table 1). Decreasing the PB capacity has a linear positive influence on cumulative pressure increase of *C. acetobutylicum*. This could be due to an accumulation of excreted organic acids and/or an increase of soluble $CO_2$ concentration with increasing cumulative gas pressure at low buffer capacity, as the low PB capacity cannot keep the pH stable. A low pH value of the medium is known to decrease the activity of [Fe-Fe]-hydrogenases of *C. acetobutylicum*, which changes the metabolic pathway from acidogenesis and acetogenesis to solventogenesis resulting in lower gas generation[46]. Moreover, it has been observed that a metabolic

**Table 1 Concentrations of the compounds in DoE media.**

| DoE buffer | Ammonium source | Acetate source | Phosphate buffer (PB) capacity | |
|---|---|---|---|---|
| | $NH_4Cl$ (AC) (mmol L$^{-1}$) | $NaCH_3COO$ (SA) (mmol L$^{-1}$) | $KH_2PO_4$ (mmol L$^{-1}$) | $K_2HPO_4$ (mmol L$^{-1}$) |
| A | 120 | 3 | 150 | 60 |
| B | 120 | 3 | 3 | 1.2 |
| C | 120 | 30 | 3 | 1.2 |
| D | 120 | 30 | 150 | 60 |
| E | 65 | 16.5 | 76.5 | 30.5 |
| F | 10 | 30 | 150 | 60 |
| G | 10 | 3 | 3 | 1.2 |
| H | 10 | 30 | 3 | 1.2 |
| I | 10 | 3 | 150 | 60 |

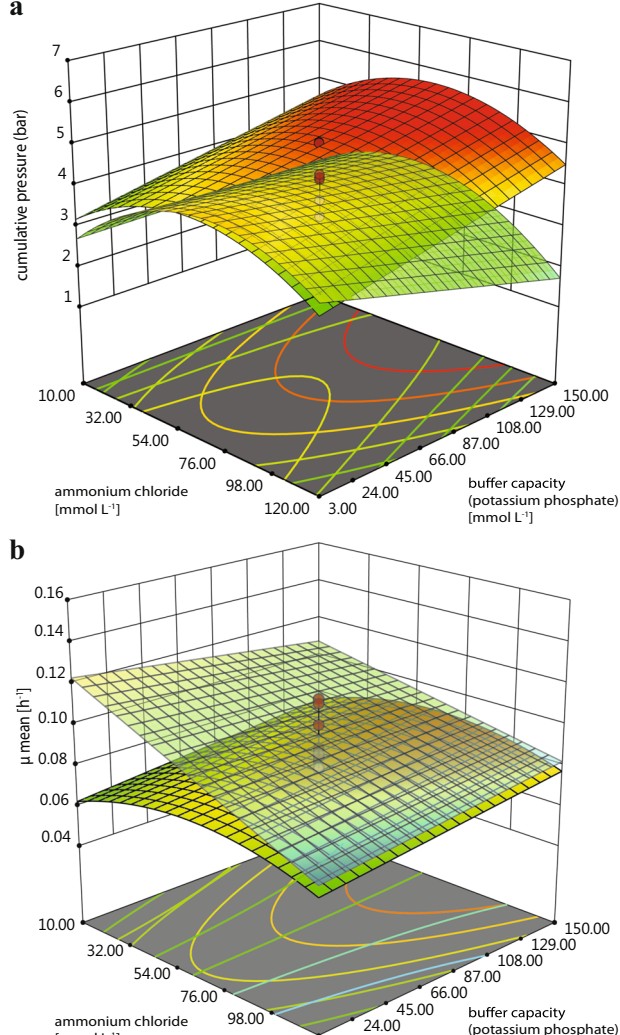

**Fig. 2 Response surface plots of *C. acetobutylicum* and *E. aerogenes* in different DoE media.** Overlay of cumulative pressure as function of ammonium chloride and buffer capacity of *C. acetobutylicum* and *E. aerogenes* is shown in **a**, and $\mu_{mean}$ as a function of ammonium chloride and buffer capacity of *C. acetobutylicum* and *E. aerogenes* is shown in **b**.

shift occurred from lactate and acetate production to butyrate production with pH increase from 5.3 to 6.3 during *Clostridium tyrobutyricum* fermentation[47]. The PB capacity is directly influencing the pH value. *C. acetobutylicum* can grow at a broad range

of pH from 4.5 to 7[48]. It has been observed that *C. acetobutylicum* grown at a pH of 4.5 had higher intracellular concentrations of acetate, butyrate and butanol compared to the culture grown at pH 6.5[49]. The model for $\mu_{mean}$ ($R^2 = 0.80$, *p*-value = <0.0001, Supplementary Table 1) of *C. acetobutylicum* indicates that increasing AC concentration has a linear negative influence on $\mu_{mean}$ (Fig. 2b). It has been reported that *E. aerogenes* strain E.82005 shifts the metabolic pathway from acid production (e.g., acetic acid production) to non-acid production (e.g., butanediol production) below a pH of 5.8, which results in a reduction of $H_2$ production[50]. This response was also observed in our model for cumulative pressure ($R^2 = 0.99$, *p*-value = <0.0001, Supplementary Table 2) of *E. aerogenes*, where an increasing PB capacity had a linear positive influence on gas generation. The model for $\mu_{mean}$ ($R^2 = 0.67$, *p*-value = <0.0001, Supplementary Table 2) of *E. aerogenes* is based on an optimum AC concentration, which is due to a quadratic model term, and on a linear dependence of the PB capacity. By examining the response surface plots of $\mu_{mean}$ and cumulative gas production by each of the organisms (compare response surface plots in Fig. 2b), optimum medium for high $\mu$ and cumulative gas production was identified to be medium E. Hereafter, all experiments were conducted with E-medium containing AC, SA and PB at concentrations of 65, 16.5 and 76.5 mmol L$^{-1}$, respectively.

**Mono-culture experiments.** Before the design of the optimum artificial consortium, quantitative PCR (qPCR) assays were developed due to the lack of morphological differences between the two species, to monitor the population dynamics by following the abundance of the individual species of *E. aerogenes* and *C. acetobutylicum* in the consortium. The correlation between qPCR reads and absolute number of cells was determined by mono-culture cell counting. Initially, mono-culture cultivations of *E. aerogenes* and *C. acetobutylicum* were conducted on newly designed E-medium containing glucose or cellobiose. Growth kinetics, byproduct formation, substrate uptake and HER of *E. aerogenes* on glucose and cellobiose are shown in Fig. 3a, b. $H_2$ production by *E. aerogenes* commenced at 40 h on glucose and on cellobiose. Moreover, in Fig. 3c, d growth kinetics, byproduct formation, substrate uptake and HER of *C. acetobutylicum* on glucose and on cellobiose, respectively, are presented. $H_2$ production by *C. acetobutylicum* started after 62 and 28 h on glucose or cellobiose, respectively. The global substrate uptake, yields of all byproducts and the mass balance analyses of the experiments are presented in Table 2. $H_2$ and $CO_2$ productivities and yields in between the time points from mono-culture cultivations are presented in Table 3. Mono-cultures of *E. aerogenes* and *C. acetobutylicum* resulted in maximum $Y_{(H2/S)}$ of 0.13 mol C-mol$^{-1}$ and 0.33 mol C-mol$^{-1}$ on glucose, and 0.04 acetic acid and ethanol were produced during the growth of

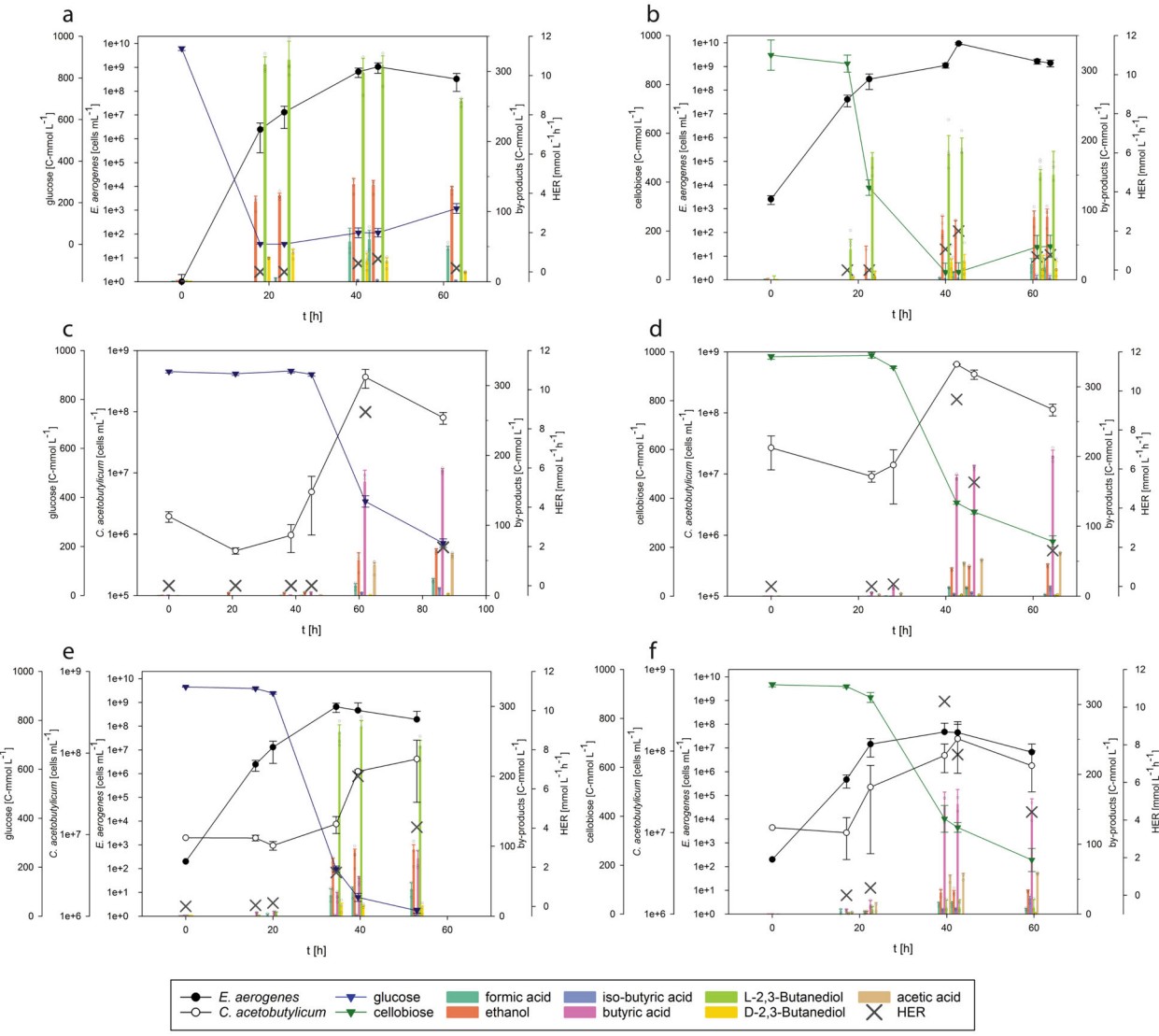

**Fig. 3 Growth, substrate uptake and production kinetics of *E. aerogenes*, *C. acetobutylicum* and the engineered consortium on glucose and cellobiose.** Growth, substrate uptake and production kinetics of *E. aerogenes* (**a**, **b**), *C. acetobutylicum* (**c**, **d**) and the consortium (with an inoculum ratio of 1:10,000 *E. aerogenes*: *C. acetobutylicum*) (**e**, **f**) on 999 C-mmol L⁻¹ glucose and cellobiose ($N = 3$ and $n = 4$). The results indicate that the amount of produced byproduct was decreased and higher HER values were reached during the in consortium cultivation compared to mono-culture on glucose and cellobiose.

*C. acetobutylicum* on glucose or cellobiose (Supplementary Table 3). In addition, at the time point where the highest HER was detected during growth of each of the mono-cultures on glucose, the community composition of mono-cultures was visualized with fluorescent in situ hybridization (FISH). The results of FISH analysis are shown in Fig. 4. *E. aerogenes* was visualized by fluorescein isothiocyanate (FITC) and tetramethylrhodamine (TRITC) signals; hence, an overlay of two probes (blue) represents *E. aerogenes*. *C. acetobutylicum* was detected by TRITC (pink signals).

**Design and experimental validation of the artificial microbial consortium.** After optimizing the substrate concentration, composing and optimizing the mutual medium, investigating the growth, substrate uptake and production kinetics of both strains, the design of the defined artificial consortium was performed with regard to the eco-physiology and biotechnological characteristics of initial ratios of microorganisms. Initial cell densities were examined with $OD_{600}$ measurements and further each time points were examined with qPCR. The first attempt was initiating the system with almost equal cell densities (1:2) of *E. aerogenes* and *C.*

*acetobutylicum*. When the initial inoculum comprised almost equal cell densities of both strains, *E. aerogenes* rapidly overgrew *C. acetobutylicum*. Therefore, the microorganisms were inoculated at different initial cell densities (*E. aerogenes* to *C. acetobutylicum* ratios of 1:100, 1:1000 and 1:10,000), and to prioritize the highest productive defined artificial consortium with respect to their growth, substrate uptake and gas production kinetics (Supplementary Fig. 3). *C. acetobutylicum* was introduced to the system directly from a pre-culture in exponential growth phase grown in E-medium, to prevent spore formation. The consortium comprising an inoculum ratio of 1:10,000 (*E. aerogenes* : *C. acetobutylicum*) showed the highest maximum HER of 6.64 mmol L⁻¹ h⁻¹ (between 34.5 and 39.5 h) and 10.3 mmol L⁻¹ h⁻¹ (between 22.5 and 39.5 h) on glucose and cellobiose, respectively (Tables 2 and 3, and Fig. 3). Furthermore, H₂ production was initiated earlier in the consortium (during the first 16 h on glucose, 22.5 h on cellobiose) compared to both mono-culture cultivations on each of the substrates (Supplementary Table 1). These findings clearly indicate that the engineered artificial microbial consortium with an inoculum ratio of 1:10,000 (*E. aerogenes* : *C. acetobutylicum*) reached higher HER

**Table 2 Global substrate uptake rate, byproduct production rates and the mass balance analyses of the mono-cultures and consortium on glucose and cellobiose.[a]**

**Glucose**

| Time [h] | Glucose uptake rate [C-mmol L⁻¹ h⁻¹] | L-2,3-Butanediol production rate [C-mmol L⁻¹ h⁻¹] | Acetic acid production rate [C-mmol L⁻¹ h⁻¹] | Formic acid production rate [C-mmol L⁻¹ h⁻¹] | Isobutyric acid production rate [C-mmol L⁻¹ h⁻¹] | Lactic acid production rate [C-mmol L⁻¹ h⁻¹] | Ethanol production rate [C-mmol L⁻¹ h⁻¹] | D-2,3-Butanediol production rate [C-mmol L⁻¹ h⁻¹] | Butyric acid production rate [C-mmol L⁻¹ h⁻¹] | Biomass (x) [C-mmol L⁻¹] | C-balance | DoR |
|---|---|---|---|---|---|---|---|---|---|---|---|---|
| *E. aerogenes* 0–63 | 12.21 ± 0.43 | 4.06 ± 0.08 | | 0.73 ± 0.08 | 0.03 ± 0.003 | 0.12 ± 0.02 | 2.09 ± 0.08 | 0.21 ± 0.02 | | 3.64 ± 0.20 | 0.96 ± 0.02 | 1.12 ± 0.02 |
| *C. acetobutylicum* 0–62 | 8.61 ± 0.65 | | 0.74 ± 0.03 | 0.23 ± 0.04 | 0.06 ± 0.02 | | 0.81 ± 0.07 | | 2.58 ± 0.32 | 3.86 ± 0.04 | 1.12 ± 0.1 | 1.03 ± 0.02 |
| Consortium 0–53 | 17.25 ± 0.05 | 3.23 ± 0.60 | 0.91 ± 0.01 | 0.80 ± 0.15 | 0.03 ± 0.02 | | 0.86 ± 0.27 | 0.21 ± 0.11 | 1.54 ± 0.20 | 2.61 ± 0.54 | 0.95 ± 0.04 | 1.13 ± 0.14 |

**Cellobiose**

| Time [h] | Cellobiose uptake rate [C-mmol L⁻¹ h⁻¹] | L-2,3-Butanediol production rate [C-mmol L⁻¹ h⁻¹] | Acetic acid production rate [C-mmol L⁻¹ h⁻¹] | Formic acid production rate [C-mmol L⁻¹ h⁻¹] | Isobutyric acid production rate [C-mmol L⁻¹ h⁻¹] | Glucose production rate [C-mmol L⁻¹ h⁻¹] | Ethanol production rate [C-mmol L⁻¹ h⁻¹] | D-2,3-Butanediol production rate [C-mmol L⁻¹ h⁻¹] | Butyric acid production rate [C-mmol L⁻¹ h⁻¹] | Biomass (x) [C-mmol L⁻¹] | C-balance | DoR |
|---|---|---|---|---|---|---|---|---|---|---|---|---|
| *E. aerogenes* 0–61 | 7.77 ± 0.7 | 2.65 ± 0.21 | | 0.36 ± 0.11 | 0.01 ± 0.001 | 0.34 ± 0.08 | 1.52 ± 0.20 | 0.27 ± 0.17 | | 1.69 ± 0.004 | 1.05 ± 0.03 | 1.12 ± 0.03 |
| *C. acetobutylicum* 0–46.6 | 13.96 ± 0.27 | | 1.11 ± 0.03 | 0.26 ± 0.02 | 3.96 ± 0.05 | 0.11 ± 0.03 | 0.90 ± 0.05 | | 0.11 ± 0.03 | 4.26 ± 0.07 | 1.02 ± 0.02 | 1.10 ± 0.02 |
| Consortium 0–42.5 | 14.18 ± 0.23 | 0.33 ± 0.10 | 1.26 ± 0.10 | 0.34 ± 0.07 | 0.18 ± 0.06 | | 0.74 ± 0.12 | | 3.69 ± 0.40 | 1.84 ± 0.30 | 1.00 ± 0.08 | 1.02 ± 0.1 |

[a]Substrate uptake rate, byproduct production rates and the mass balance analyses were conducted between the first and last time points (global).

values on both substrates (Fig. 3) compared to the other inoculum ratios. When the 1:10,000 mixing ratio was employed, the quantities of excreted liquid metabolic byproducts were also decreased (Fig. 3). The consortium displayed 1.24-fold lower butanediol production compared to mono-culture of *E. aerogenes* and 1.57-fold lower butyric acid production compared to mono-culture of *C. acetobutylicum* on glucose. Lower amounts of ethanol and higher amounts of acetic acid and formic acid production were also detected during our consortium experiments (Table 2). Validity of the $H_2$ production and productivities of the byproducts was also confirmed by calculating the C- and degree of reduction (DoR) balances. Global byproduct formation rates, substrate uptake, the mass balance analyses of all experiments and growth kinetics are shown in Table 2. The substrate uptake rate was higher in the consortium experiments compared to mono-culture experiments, for both substrates. HER, yields of gases ($Y_{(H2/S)}$, yield of $CO_2$ per substrate consumed ($Y_{(CO2/S)}$)) and $qH_2$ are shown in Table 3 for each time point. The optimum consortium comprising an inoculum ratio of 1:10,000 (*E. aerogenes*:*C. acetobutylicum*) showed $Y_{(H2/S)}$ of 0.93 $mol_{(H2)}$ C-mol⁻¹ on glucose (between 39.5 and 53 h), which is equal to 5.58 $mol_{(H2)}$ mol⁻¹, and 0.73 $mol_{(H2)}$ C-mol⁻¹ (4.38 $mol_{(H2)}$ mol⁻¹$_{(C6\ sugar\text{-}equivalent)}$) on cellobiose (between 39.5 and 42.5 h) (Table 3 and Supplementary Table 4). To our knowledge, this is the first study that describes an improvement of $Y_{(H2/S)}$ beyond the Thauer limit in defined medium without automated gas removal techniques. These results indicate that precision design of substrate concentration, medium compounds, activity and ratio of organisms must be fine-tuned to meet the eco-physiological prerequisites of the utilized organisms to improve substrate uptake, growth and production kinetics clearly beyond the reported values.

Then we performed FISH to snapshot the population composition and visualize the interaction during the cultivation of the consortium on glucose. Both *E. aerogenes* (blue) and *C. acetobutylicum* (pink) were visualized for each sample taken from different time points (from 0 h (time point zero, after inoculation) to 53 h (time point 5)) (Supplementary Fig. 4). FISH confirmed that intact cells of both microorganisms were contributing to the artificial microbial community and verified the homogenous distribution of microorganisms, which was obtained by qPCR (Fig. 3). At these time points, ecological indicators were also assessed. From the Shannon Index (H) and species richness (S), the evenness ($E_H$) was calculated (Supplementary Table 5). The results indicate that the microbial community was almost evenly distributed during the time point of maximum $Y_{(H2/S)}$ on cellobiose ($E_H = 0.79$) and slightly less diverse at the maximum $Y_{(H2/S)}$ on glucose ($E_H = 0.51$).

## Discussion

Renewably produced $H_2$ could be implemented as one of the main energy carriers of the twenty-first century[51]. To gain biological $H_2$ production at the theoretical $Y_{(H2/S)}$, different methods (e.g., reactor configurations[52], metabolic engineering[53], modelling and optimization[54], statistical analysis[33], pre-treatment strategies for spore germination, nutrient formulations, substrate composition and concentration[55]) were proposed and/or already investigated. Using $H_2$-producing defined or undefined consortia was considered as one of the auspicious approaches[38]. However, an undefined consortium fetches many technical problems due to the reaction complexity, process kinetics, difficulties of optimization and various process parameters (e.g., pH and temperature), as well as the ecological and functional aspects of the system[10]. Furthermore, $H_2$ formation is not the prime aim of microbes, but the microorganism aims on optimizing the energy yield. These two aspects might be in conflict to a certain extent, but a defined consortium allows better control regarding $H_2$ formation,

**Table 3 Productivities and $Y_{(H2/S)}$ of the mono-cultures and consortium grown on glucose and cellobiose[a].**

**Glucose**

| Time [h] | $Y_{(CO2/S)}$ [mol C-mol$^{-1}$] | $Y_{(H2/S)}$ [mol C-mol$^{-1}$] | HER [mmol L$^{-1}$ h$^{-1}$] | qH$_2$ [mmol h$^{-1}$ g$^{-1}$] | CER [mmol L$^{-1}$ h$^{-1}$] |
|---|---|---|---|---|---|
| *E. aerogenes* | | | | | |
| 0-18 | | | | | |
| 0-23.5 | | | | | |
| 0-40.5 | 0.03 ± 0.003 | 0.02 ± 0.002 | 0.43 ± 0.14 | | 0.61 ± 0.13 |
| 40.5-45 | 0.23 ± 0.06 | 0.14 ± 0.04 | 0.66 ± 0.32 | 11.34 ± 0.65 | 1.18 ± 0.46 |
| 45-63 | 0.10 ± 0.03 | 0.05 ± 0.02 | 0.19 ± 0.05 | | 0.35 ± 0.08 |
| *C. acetobutylicum* | | | | | |
| 0-21 | | | | | |
| 0-38.5 | | | | | |
| 0-45 | | | | | |
| 45-62 | 0.17 ± 0.03 | 0.30 ± 0.06 | 8.86 ± 0.60 | 86.41 ± 4.51 | 5.03 ± 0.7 |
| 62-86.5 | 0.46 ± 0.17 | 0.33 ± 0.12 | 1.96 ± 0.84 | | 2.75 ± 0.50 |
| Consortium | | | | | |
| 0-16.0 | 0.09 ± 0.01 | 0.12 ± 0.05 | 0.05 ± 0.01 | | 0.06 ± 0.01 |
| 16-20.0 | 0.08 ± 0.05 | 0.05 ± 0.03 | 0.17 ± 0.02 | 68.54 ± 5.4 | 0.27 ± 0.01 |
| 20-34.5 | 0.09 ± 0.01 | 0.04 ± 0.01 | 1.73 ± 0.5 | 9.60 ± 3.2 | 4.15 ± 0.5 |
| 34.5-39.5 | 0.43 ± 0.02 | 0.44 ± 0.1 | 6.64 ± 0.25 | 213.98 ± 8.8 | 16.19 ± 0.4 |
| 39.5-53.0 | 0.67 ± 0.28 | 0.93 ± 0.29 | 4.04 ± 0.22 | | 5.84 ± 0.21 |
| **Cellobiose** | | | | | |
| Time [h] | $Y_{(CO2/S)}$ [mol C-mol$^{-1}$] | $Y_{(H2/S)}$ [mol C-mol$^{-1}$] | HER [mmol L$^{-1}$ h$^{-1}$] | qH$_2$ [mmol h$^{-1}$ g$^{-1}$] | CER [mmol L$^{-1}$ h$^{-1}$] |
| *E. aerogenes* | | | | | |
| 0-17.5 | | | | | |
| 0-22.5 | | | | | |
| 0-40 | 0.08 ± 0.01 | 0.04 ± 0.003 | 1.07 ± 0.05 | 4.68 ± 0.91 | 2.44 ± 0.23 |
| 40-43 | 0.09 ± 0.01 | 0.02 ± 0.002 | 2.00 ± 0.3 | 0.88 ± 0.71 | 9.35 ± 0.61 |
| 43-61 | 0.18 ± 0.02 | 0.04 ± 0.004 | 0.67 ± 0.06 | | 2.96 ± 0.19 |
| 61-64 | 0.06 ± 0.01 | 0.01 ± 0.002 | 0.78 ± 0.03 | | 3.30 ± 0.24 |
| *C. acetobutylicum* | | | | | |
| 0-23 | | | | | |
| 0-28 | 0.01 ± 0.002 | 0.01 ± 0.002 | 0.11 ± 0.04 | 83.43 ± 4.31 | 0.10 ± 0.03 |
| 28-42.5 | 0.14 ± 0.001 | 0.25 ± 0.002 | 9.57 ± 0.13 | 55.78 ± 9.74 | 5.43 ± 0.10 |
| 42.5-46.5 | 0.38 ± 0.01 | 0.43 ± 0.01 | 5.32 ± 0.31 | | 7.64 ± 0.25 |
| 46.5-64.5 | 0.41 ± 0.02 | 0.27 ± 0.03 | 1.83 ± 0.12 | | 2.80 ± 0.13 |
| Consortium | | | | | |
| 0-17 | | | | | |
| 0-22.5 | 0.01 ± 0.01 | 0.06 ± 0.03 | 0.39 ± 0.01 | 34.77 ± 3.5 | 0.06 ± 0.01 |
| 22.5-39.5 | 0.12 ± 0.02 | 0.36 ± 0.05 | 10.31 ± 0.22 | 433.95 ± 10.1 | 3.32 ± 0.31 |
| 39.5-42.5 | 0.72 ± 0.11 | 0.73 ± 0.18 | 7.48 ± 0.16 | 528.81 ± 8.11 | 7.30 ±.013 |
| 45.5-59.5 | 0.13 ± 0.06 | 0.63 ± 0.28 | 4.42 ± 0.20 | | 0.94 ± 0.32 |

[a]Values were calculated in between time points (after gas production started) as indicated in the table.

whereas an undefined mix of microorganisms will tend to optimize energy formation. Therefore, an artificial/defined consortium, with well-studied microorganisms, is essential to further understand the relationship among microorganisms and to allow sophisticated process control, as the physiologies of the members of microbial community can be examined in depth and individually as well as mutually optimized. So far, artificial dark fermentative H$_2$-producing consortia were utilized in over 40 studies, which we summarized with respect to dark fermentative H$_2$ production and their main parameters (Supplementary Data 1). The highest reported $Y_{(H2/S)}$ was 4.42 mol$_{(H2)}$ mol$^{-1}$$_{(glucose)}$, which corresponds to 0.74 mol$_{(H2)}$ C-mol$^{-1}$) from a consortium of *Caldicellulosiruptor saccharolyticus* and *Caldicellulosiruptor owensensis*[56], followed by a thermophilic consortium composed of *C. saccharolyticus* and *Caldicellulosiruptor kristjanssonii* comprising 3.8 mol$_{(H2)}$ mol$^{-1}$$_{(C6 sugar-equivalent)}$ (0.63 mol$_{(H2)}$ C-mol$^{-1}$)[29]. Both studies were conducted on complex medium containing yeast extract. The highest $Y_{(H2/S)}$ reported from a mesophilic consortium of *Enterobacter cloacae* and *Bacillus cereus*, was a $Y_{(H2/S)}$ of 3 mol$_{(H2)}$ mol$^{-1}$$_{(glucose)}$, which is the equivalent to 0.5 mol$_{(H2)}$ C-mol$^{-1}$[57], followed by a consortium of *E. aerogenes* and *Clostridium butylicum*, with a $Y_{(H2/S)}$ of 2.7 mol$_{(H2)}$ mol$^{-1}$$_{(glucose)}$ (0.45 mol$_{(H2)}$ C-mol$^{-1}$)[58].

Our study is the first of its kind, which considered and integrated results from several physiological, ecological and biotechnological levels: (1) meta-data analysis and modelling pipeline of dark fermentative H$_2$ producers[32]; (2) physiological, ecological and biotechnological aspects of mono- and co-culture design; (3) optimization of H$_2$ production by subsequently investigating the effect of substrate concentration on growth and gas production; (4) employing DoE method to design a mutual defined medium (E-medium); and, finally, (5) engineering a defined artificial consortium by examining different initial ratios of microorganisms in defined medium. Here, we present an optimum consortium comprising two species with an inoculum ratio of 1 : 10,000 (*E. aerogenes* : *C. acetobutylicum*) with a $Y_{(H2/S)}$ of 5.58 mol$_{(H2)}$ mol$^{-1}$$_{(glucose)}$ (0.93 mol$_{(H2)}$ C-mol$^{-1}$) and 4.38 mol$_{(H2)}$ mol$^{-1}$$_{(C6 sugar-equivalent)}$ (0.73 mol$_{(H2)}$ C-mol$^{-1}$) on glucose and cellobiose, respectively. This precisely engineered consortium comprised the highest ever reported $Y_{(H2/S)}$ and clearly surpassed the Thauer limit. Our findings point at a yet unidentified synergistic effect of the two strains that improves H$_2$ production.

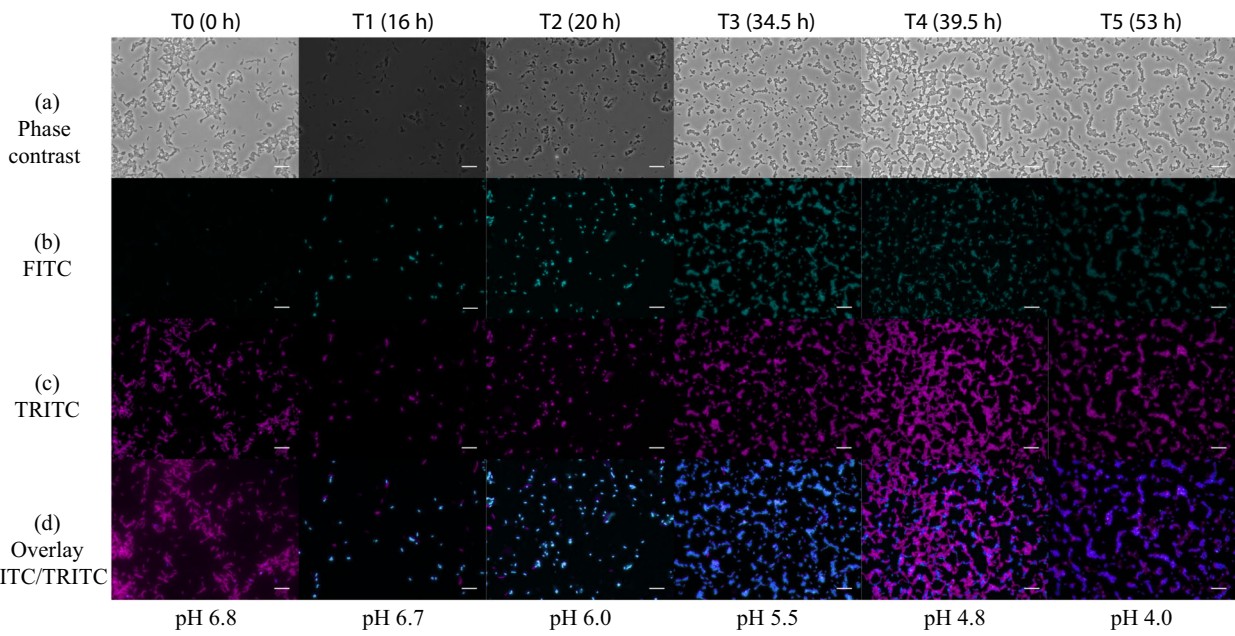

**Fig. 4 Fluorescent in situ hybridization (FISH) analysis of _E. aerogenes_ and _C. acetobutylicum_ mono-cultures on glucose and the consortium on glucose and cellobiose.** The consortium comprised an inoculum ratio of 1:10,000 _E. aerogenes_: _C. acetobutylicum_. The phase-contrast images (**a**), FITC filter set images (**b**), TRITC filter set images (**c**) and images from overlay of FITC/TRITC filter sets (**d**) are shown at the time point where gas production was the highest (T4). The scale bar is 10 μm.

The E-medium composition had a major effect on the metabolism of the microorganisms. The obtained metabolic byproducts highlight the active metabolic routes of the microorganisms. On _Enterobacter_-specific medium, we showed that byproducts of _E. aerogenes_ were mainly acetate and ethanol[59]. In our study, the _E. aerogenes_ mono-culture produced high amounts of 2,3-butanediol, which is an industrial chemical and liquid fuel and is used in food, cosmetics and medicine industries[60,61]. It has been reported that 2,3-butanediol is produced by _E. aerogenes_ under molecular oxygen-limiting and anaerobic conditions[62,63], and that the initial acetate source induces butanediol production by catalysing the breakdown of pyruvate to butanediol[64]. E-medium contains acetate and that might be the reason of production of this compound. In addition, a higher level of $CO_2$ was observed during mono-culture _E. aerogenes_ cultivation, which is again confirming butanediol fermentation. Production of 2,3-butanediol could not be detected during _C. acetobutylicum_ mono-culture cultivation (Fig. 3 and Table 2). Compared to the mono-culture experiments, it was observed that during the consortium experiments the release of metabolic end products of the two species changed. Lower amounts of 2,3-butanediol were also detected during the consortium cultivation on glucose compared to mono-culture of _E. aerogenes_ (Fig. 3 and Table 2). This is another indication of an operative consortium where both members were metabolically functional. Moreover, during the consortium experiments production of acetic acid was higher and ethanol production was decreased, which most likely provided room for $H_2$ production. Another aspect of the precision design of the medium was the PB capacity. At pH of 5.5, the consortium was able to produce $H_2$ due to the activity of _C. acetobutylicum_. In biohydrogen production, pH values < 4.5 lead to changes in the metabolic pathways towards decreased concentrations of undissociated forms of organic acids, which cause possible inhibition of hydrogenase activity[65,66], affecting ferredoxins' capacity to donate electrons to reduce protons[66,67] and affect microbial growth[66,68]. Low pH also induces the sporulation of _C. acetobutylicum_, which can be observed in the last time point of our FISH images in

Supplementary Fig. 4. The concentration of _C. acetobutylicum_ (coloured pink) at the time point 5 was drastically decreased at the FISH image (Supplementary Fig. 4) and the qPCR reads (Fig. 3).

Another investigated aspect in this study was the initial cell densities of each microorganism. In previous consortium studies, and most of the cases, an equal suspension volume with an unknown amount of living/active cells of each organism has been used for inoculation[29,30]. To our knowledge, this is the first study in which consortium was engineered by introducing active microorganisms at initial cell densities of five orders of magnitude difference into the system. The functional co-existence of two bacteria was shown, when they were introduced to the system at all aforementioned inoculum ratios. Expectedly, cell densities of microorganisms and gas production values differed at each inoculum ratio (Supplementary Fig. 3). This was an additional indication of the importance for precision design of the consortium including biotic and abiotic factors. Furthermore, $H_2$ production was initiated earlier in the consortium (during the first 16 h on glucose, 22.5 h on cellobiose) compared to both mono-culture cultivations on each of the substrates. These findings clearly indicate that the engineered consortium with an inoculum ratio of 1:10,000 (_E. aerogenes_ : _C. acetobutylicum_) reached higher HER values on different substrates; thus, $H_2$ production kinetics are superior over mono-cultures (Fig. 3).

This study presents an interdisciplinary approach to improve $H_2$ production beyond the Thauer limit from the molecular to the process level, and enlightens a systematic and engineering understanding and description of the kinetic and mechanistic aspects, which are responsible for design and definition of this efficient artificial microbial consortium. Constructing the consortium with this approach could also improve the productivity/ yield of natural or undefined consortia and provide controllable, stable, predictable biotechnological processes over currently existing systems. Precision design of microbial communities might be employed for the targeted enrichment of microorganisms in undefined microbial populations or for the restoration of

microbial ecosystems in plant, animal and human health, or in bioremediation. Design of synthetic microbial communities for the targeted conversion of complex biopolymers or surplus electricity to biofuels or intermediate storage molecules such as formic acid will benefit from the specific development of communities of well-characterized pure cultures with known growth, substrate uptake and production kinetics, which are aligned by selecting appropriate concentrations of substrate, pH, reduction potential, salt concentration, inoculum size and co-substrate availability, or the mutual exchange of metabolic byproducts between the syntrophic partners in the synthetic microbial community.

The present study is a major leap forward in the design of an artificial microbial consortium through precision engineering. Our improvement route is unprecedented and delivers an active, balanced and highly functional co-existence of two bacteria with improved $H_2$ production kinetics. The $H_2$ production characteristics of this defined artificial consortium is superior compared to any mono-, co- or multi-culture system reported to date. The system could be further improved to enhance $H_2$ production by introducing other microorganisms into the consortium, and the stability of the system can be boosted by $H_2$ milking technology[69,70], or can be combined with methanogenic archaea to stimulate syntrophic growth. Moreover, precision design of an artificial microbial consortium could even serve as a template for conversion of cellulosic biomass to gaseous and liquid biofuels. Our blueprint for a precision design consortium could hence be further extended for the development of consolidated bioprocesses for targeted conversion of lignocellulosic biomass to liquid biofuels, for the development of start-up communities in anaerobic digestion, for the conversion complex gas mixtures, food waste utilization or (bio)plastic recycling.

In conclusion, the precision engineered consortium exhibited highly efficient $H_2$ production from glucose and cellobiose compared to the mono-cultures of either microorganism under optimal conditions or compared to any consortium reported in literature. Our drawing board-like design of a defined artificial microbial consortium of microorganisms improved HER beyond reported values. The engineered consortium breaking the Thauer limit displayed 6.6- and 2.8-fold higher maximum $Y_{(H2/S)}$ on glucose and 18.3- and 1.7-fold higher maximum $Y_{(H2/S)}$ on cellobiose compared to mono-cultures of E. aerogenes and C. acetobutylicum, respectively. The precision design of artificial microbial consortia, which considers results from a priori physiological and biotechnological knowledge from meta-data analysis will lead to a breakthrough in biotechnology by improving productivity and yield. However, this study indicates that the precision design of artificial microbial consortia might only be efficacious when nutrient demands of the individual members are individually and mutually aligned with the eco-physiological characteristics of the organisms. The eco-physiological requirements of microorganisms in undefined ecosystems have to be considered at a strain level to be able to improve the performance of the individual players in the community and to achieve high production rates and yields.

## Methods

**Microorganisms and medium composition.** C. acetobutylicum DSM 792 and E. aerogenes DSM 30053 were used for all experiments. A modified Clostridium-specific medium without yeast extract was used for growth of mono-culture C. acetobutylicum as previously described in detail elsewhere[71]. The medium was prepared containing (per L): 0.5 g of $KH_2PO_4$, 0.5 g of $K_2HPO_4$ and 2.2 g of $NH_4CH_3COO$ and glucose or cellobiose were added at a concentration of 999 C-mmol. The pH was arranged with 1 mol $L^{-1}$ NaOH to 6.8. Trace elements solution was prepared as stock 100× solution containing (per L): 0.2 g of $MgSO_4·7 H_2O$, 0.01 g of $MnSO_4·7H_2O$, 0.01 g of $FeSO_4·7H_2O$, 0.01 g of NaCl. Vitamin solution was prepared as stock 200× solution containing (per L): 0.9 g of thiamine, 0.002 g

of biotin and 0.2 g of 4-aminobenzoic acid. The trace elements solution and the vitamin solution were used for all experiments. Mono-culture of E. aerogenes was grown in a defined Enterobacter-specific medium, as described elsewhere[72]. The Enterobacter-specific medium was prepared containing (per L): 13.3 g $K_2HPO_4$, 4 g $(NH_4)_2HPO_4$, 8 mg EDTA and trace elements (2.5 mg $CoCl_2·6H_2O$, 15 mg $MnCl_2·4H_2O$, 1.5 g $CuCl_2·4H_2O$; 3 mg $H_3BO_3$; 2.5 mg $Na_2MoO_4·2H_2O$, 13 mg of $Zn(CH_3COO)_2·2H_2O$). Glucose and cellobiose were prepared as stock solutions. Media, trace element solution, glucose and cellobiose solutions were flushed with sterile $N_2$ to make the solutions anaerobic and sterilized separately at 121 °C for 20 min. Sterile anaerobic solutions of glucose or cellobiose, trace elements solution and filter sterilized vitamin solution were added into the media before the inoculation inside the sterilized biological safety cabinet (BH-EN 2005, Faster Srl, Ferrara, Italy).

**Design of experiments.** A mutual medium accommodating the nutritional requirements of both organisms was designed by using the DoE approach. The buffer compositions of two species specific media described above were analysed and the optimum concentrations of AC ($NH_4Cl$), SA ($Na^+$ acetate) and PB ($KH_2PO_4/K_2HPO_4$) capacity were investigated. The setting of DoE for concentration effect of AC, SA and PB capacity was based on 29 randomized runs within concentration range from 3–30 mmol $L^{-1}$ of AC, 3–150 mmol $L^{-1}$ of $KH_2PO_4$ and 10–120 mmol $L^{-1}$ of SA (Table 1). Each experiment was performed in triplicates ($n = 3$), except for set E of the DoE experiment (centre points), which were performed in pentaplicate ($n = 5$). The DoE experiments were performed twice ($N = 2$). The end of the exponential growth phase of E. aerogenes and C. acetobutylicum was reached at 45 and 51.5 h, respectively. For modelling, these time points were used. The reason for providing an acetate source in the medium was due to the possibility to add an acetate oxidizing microorganism to the co-culture consortium, which was not performed in the context of this study.

**Closed batch cultivations.** Cultures of E. aerogenes and C. acetobutylicum were grown anaerobically at 0.3 bar in a 100 Vol.-% $N_2$ atmosphere in a closed batch set-up[33]. Mono-culture and consortium closed batch experiments were conducted with the final volume of 50 mL medium in 120 mL serum bottles (Ochs Glasgerätebau, Langerwehe, Germany). Each serum bottle contained 45 mL Clostridium-specific medium, Enterobacter-specific medium or E-medium, 0.25 mL vitamin solution, 3.0 mL glucose or cellobiose stock solution, 0.5 mL trace elements solution and 1.25 mL inoculum. The serum bottles were sealed with rubber stoppers (20 mm butyl ruber, Chemglass Life Science LLC, Vineland, USA). For consortium experiments, different inoculum ratios were tested and initial cell concentrations were arranged with the ratios of (E. aerogenes : C. acetobutylicum) 1:2, 1:10, 1:100, 1:1000, 1:10,000 and 1:100,000 at a temperature of 37 °C. Pre-culture of E. aerogenes was diluted in DoE E-medium (Table 1) to inoculate the organism at cell densities of aforementioned ratios. The pressure in the headspace of the serum bottles were measured individually using a manometer (digital manometer LEO1-Ei,−1…3 bar, Keller, Germany). After each measurement, the pressure was released completely from the headspace of serum bottle by penetrating the butyl rubber stopper with a sterile needle. The pressure values were added up to reveal total produced pressure (cumulative pressure). Experiments were performed three times ($N = 3$) and each set was performed in quadruplicates ($n = 4$).

**Cell counting, absorption measurements, DNA extraction and qPCR.** A volume of 1 mL of liquid sample was collected by using sterile syringes at regular intervals for monitoring biomass growth by measuring the absorbance (optical density at 600 nm ($OD_{600}$)) using a spectrophotometer (Beckman Coulter Fullerton, CA, USA). Every sampling operation was done inside the sterilized biological safety cabinet (BH-EN 2005, Faster Srl, Ferrara, Italy).

E. aerogenes and C. acetobutylicum cells were counted using a Nikon Eclipse 50i microscope (Nikon, Amsterdam, Netherlands) at each liquid/biomass sampling point. The samples for cell count were taken from each individual closed batch run using syringes (Soft-Ject, Henke Sass Wolf, Tuttlingen, Germany) and hypodermic needles (Sterican size 14, B. Braun, Melsungen, Germany). Ten microlitres of sample were applied onto a Neubauer improved cell counting chamber (Superior Marienfeld, Lauda-Königshofen, Germany) with a grid depth of 0.1 mm.

DNA for qPCR was extracted from 1 mL culture samples by centrifugation at 4 °C and 13,400 r.p.m. for 30 min. The following steps were applied for DNA extraction; (1) cells were resuspended in pre-warmed (65 °C) 1% sodium dodecyl sulfate (SDS) extraction buffer and (2) transferred to Lysing Matrix E tubes (MP Biomedicals, Santa Ana, CA, USA) containing an equal volume of phenol/chloroform/isoamylalcohol (25:24:1). (3) Cell lysis was performed in a FastPrep-24 (MP Biomedicals, NY, USA) device with speed setting 4 for 30 s and the lysate was centrifuged at 13,400 r.p.m. for 10 min. (4) An equal volume of chloroform/isoamylalcohol (24:1) was added to the supernatant of the lysate, followed by centrifugation at 13,400 r.p.m. for 10 min and collection of the aqueous phase. (5) Nucleic acids were precipitated with double volume of polyethylenglycol (PEG) solution (30% PEG, 1.6 mol $L^{-1}$ NaCl) and 1 μL glycogen (20 mg $mL^{-1}$) as carrier, incubated for 2 h at room temperature. (6) Following centrifugation at 13,400 r.p.m. for 1 h, nucleic acid pellets were washed with 1 mL cold 70% ethanol, dried at 30 °C using a SpeedVac centrifuge (Thermo Scientific, Dreieich, Germany), eluted in

Tris-EDTA buffer and stored at −20 °C until further analysis. Nucleic acid quantification was performed with NanoDrop ND-1000 spectrophotometer (NanoDrop Technologies, Wilmington, DE, USA). qPCR assays were developed for quantifying *E. aerogenes* and *C. acetobutylicum* in consortium. The primer pairs were designed by targeting species specific genes (Supplementary Table 6) to prevent false-positive amplification and sequences of genes were compared for identifying optimal primer using the ClustalW2 multiple sequence alignment programme (http://www.ebi.ac.uk/Tools/clustalw2/). qPCR assays were performed in Eppendorf Mastercycler epgradientS realplex[2] (Eppendorf, Hamburg, Germany). The PCR mixture (20 μL) contained 10 μL SYBR Green labelled Luna Universal qPCR Master Mix (M3003L, New England Biolabs), 0.5 μL of forward and 0.5 μL reverse primer, 8 μL sterile DEPC water and 1 μL of DNA template. Negative controls containing sterile diethyl pyrocarbonate (DEPC) water as a replacement for the DNA templates and DNA template of the non-targeted species were included separately in each run. The amplification protocol started with an initial denaturation at 95 °C for 2 min, followed by 45 cycles of denaturation at 95 °C for 30 s, annealing and fluorescence acquisition at 60 °C for 30 s and elongation at 72 °C for 30 s. A melting-curve analysis (from 60 °C to 95 °C at a transition rate of 1 °C every 10 s) was performed to determine the specificity of the amplification. All amplification reactions were performed in triplicates. A standard curve was generated as described elsewhere[29]. Culture samples of each organism were collected at different time intervals for cell count and genomic DNA extraction cell density of each strain were determined by cell counting under microscope during growth and subsequent gDNA extraction was applied to reflect absolute quantification. Six tenfold dilution standards were prepared and a linear regression analysis was performed between qPCR reads and cell counts and $OD_{600}$ measurements.

**Quantification of gas composition.** Gas chromatography (GC) measurements were performed from serum bottles that remained without any manipulation after inoculation until the first time point GC measurement. After every GC measurement, remaining gas was released completely from the serum bottles by penetrating the butyl rubber stopper using a sterile needle. The pressure of serum bottles headspace was determined to examine whether there was any remaining over-pressure by using a manometer (digital manometer LEO1-Ei,−1...3 bar, Keller, Germany). The gas compositions were analysed by using a GC (7890 A GC System, Agilent Technologies, Santa Clara, USA) with a 19808 Shin Carbon ST Micro-packed Column (Restek GmbH, Bad Homburg, Germany) and provided with a gas injection and control unit (Joint Analytical System GmbH, Moers, Germany) as described before[73–75]. The standard test gas employed in GC comprised the following composition: 0.01 Vol.-% $CH_4$; 0.08 Vol.-% $CO_2$ in $N_2$ (Messer GmbH, Wien, Austria). All chemicals were of highest grade available. $H_2$, $CO_2$, $N_2$, 20 Vol.-% $H_2$ in $CO_2$ and 20 Vol.-% $CO_2$ in $N_2$ were of test gas quality (Air Liquide, Schwechat, Austria).

**Quantification of liquid metabolites.** Quantification of sugars, volatile fatty acids and alcohols were performed with high-performance liquid chromatography (HPLC) system (Agilent 1100), consisting of a G1310A isocratic pump, a G1313A ALS autosampler, a Transgenomic ICSep ICE-ION-300 column, a G1316A column thermostat set at 45 °C and a G1362A RID refractive index detector, measuring at 45 °C (all modules were from Agilent 1100 (Agilent Technologies, CA, USA). The measurement was performed with 0.005 mol $L^{-1}$ $H_2SO_4$ as solvent, with a flow rate of 0.325 mL min$^{-1}$ and a pressure of 48–49 bar. The injection volume was 40 μL.

**Data analysis.** For the quantitative analysis, the maximum specific growth rate ($\mu_{max}$ [h$^{-1}$]) and mean specific growth rate ($\mu_{mean}$ [h$^{-1}$]) were calculated as follows: $N = N^0 \cdot e^{\mu t}$ with $N$, cell number [cells ml$^{-1}$]; $N^0$, initial cell number [cells ml$^{-1}$]; $t$, time [h] and $e$, Euler's number. According to the delta cell counts in between sample points, $\mu$ was assessed. The $Y_{(H2/S)}$ [mol mol$^{-1}$], HER [mmol $L^{-1}$ h$^{-1}$], CER [mmol $L^{-1}$ h$^{-1}$] and the specific $H_2$ production rate (q$H_2$) [mmol g$^{-1}$ h$^{-1}$][32] were calculated from the intervals between each time point and the gas composition in the headspace of serum bottle was determined using the GC. The elementary composition of the corresponding biomass[59] was used for the calculation of the mean molar weight, carbon balance and the DoR balance. Yields of byproducts were determined after HPLC measurement. Values were normalized according to the zero control. Moreover, the Shannon diversity index (H) was calculated to interpret the changes in microbial diversity, accounting for both richness (S), the number of species present and abundance of different species. Relative abundance of two species was evaluated according to the calculated evenness ($E_H$) values[76]. Global substrate uptake rate, byproduct production rates and the mass balance analyses of the mono-cultures and consortium on glucose and cellobiose were calculated between the first and last time point.

**Fluorescence in situ hybridization.** For FISH, samples of 2 mL were collected for cell fixation. The samples were centrifuged in micro-centrifuge (5415-R, Eppendorf, Hamburg, Germany) for 10 min at 13,200 r.p.m. and pellets were resuspended in 0.5 mL phosphate-buffered saline (PBS) (10 mmol $L^{-1}$ of $Na_2HPO_4$/$NaH_2PO$, 130 mmol $L^{-1}$ of NaCl, pH of 7.2–7.4). After repeating this procedure twice, 0.5 mL ice-cold absolute ethanol was added to the 0.5 mL PBS/cell mixture. The

ethanol fixed samples were thoroughly mixed and then stored at −20 °C. Poly-L-lysine solution (0.01 % (v/v)) was used for coating the microscope slides (76 × 26 × 1 mm, Marienfeld-Superior, Lauda-Königshofen, Germany) containing ten reaction wells separated by an epoxy layer. After dipping the slide into the solution for 5 min, residual poly-L-lysine from the slides was removed by draining the well, followed by air-drying for several minutes. Cells were immobilized on prepared slides by adding samples (1–10 μL) on each well and air-drying. For cell dehydration, the slides were impregnated with ethanol concentrations of 50% (v/v), 80% (v/v) and 96% (v/v), respectively. The slides were dipped into each solution for 3 min, starting from the lowest concentration.

The EUB338 probe[77] was used to target specific 16S rRNA found in almost all organisms belonging to the domain of bacteria[78]. The GAM42a probe specifically binds to target regions of gammaproteobacterial 23S rRNA[79] (Supplementary Table 7). Both probes were diluted with DEPC water to a certain extent depending on the fluorescence label. Cy3-labelled EUB338 was diluted to a probe concentration of 30 ng DNA μL$^{-1}$, whereas FLUOS-labelled GAM42a was adjusted to a final concentration of 50 ng DNA μL$^{-1}$. For hybridization of the probe, 20 μL of hybridization buffer (900 mmol $L^{-1}$ NaCl, 20 mmol $L^{-1}$ Tris/HCl, 30% formamide (v/v), 0.01% SDS (v/v)) and 2 μL of diluted probe solution were added into each well. The hybridization reaction (46 °C, overnight) was facilitated using an airtight hybridization chamber (50 mL centrifuge tube) to prevent dehydration.

A stringent washing step was performed at 48 °C for 10 min in pre-warmed 50 mL washing buffer (100 mmol $L^{-1}$ NaCl, 20 mmol $L^{-1}$ Tris/HCl, 5 mmol $L^{-1}$ EDTA). Afterwards, the slides were dried up and a mounting medium (Antifade Mounting Medium, Vectashield Vector Laboratories, CA, USA) was added to each well. The slides were sealed with a cover glass and examined under phase-contrast microscope (Nikon Eclipse Ni equipped with Lumen 200 Fluorescence Illumination Systems) using filter sets TRITC (557/576) (maximum excitation/emission in nm) for cy3-labelled EUB338 probe and FITC (490/525) for FLUOS-labelled GAM42a probes by a 100 × 1.45 numerical aperture microscope objective (CFI Plan Apo Lambda DM ×100 Oil; Nikon Corp., Japan).

**Statistics and reproducibility.** DoE experiments were designed and analysed using Design Expert version 11.1.2.0 (Stat-Ease, Inc. USA). Analysis of variation was performed at $\alpha = 0.05$. The $p$-values for each test are indicated in the 'Results' section. All closed batch experiments were reproduced three times ($N = 3$) and each replication contained quadruplicate ($n = 4$). qPCR and FISH experiments, which applied all of the mentioned replicates, were performed in technical triplicates ($n = 3$). DoE experiments were conducted twice ($N = 2$) and each replication contained triplicate experiments for corner points ($n = 3$), except the set E (centre points), which were performed in biological pentaplicates ($n = 5$).

**Reporting summary.** Further information on research design is available in the Nature Research Reporting Summary linked to this article.

## Data availability
Primer and probe sequences, and full results of statistical analyses are provided in the Supplementary Information file. Primary data can be provided upon request by the corresponding author. Data are archived on local and network-based applications. Primary data are accessible via the cloud-based University of Vienna storage systems at any time on reasonable request.

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

## Acknowledgements

The Austrian Research Promotion Agency (Forschungsförderungsgesellschaft (FFG)) is gratefully acknowledged for supporting this research in the frame of the projects H2.AT (grant 853618) and NitroFix (grant 859293). The BMBWF is acknowledged for supporting the research with the WTZ project CZ 08/2020. We thank Philipp Weber, MSc, BSc, and Kevin Pfeifer, MSc, BSc, for their help with FISH and imaging analyses. Open access funding provided by the University of Vienna.

## Author contributions

İ.E. and S.K.-M.R.R. conceived and planned the experiments, and wrote the manuscript. İ.E. carried out the data analysis and calculations. İ.E. designed and optimized all the methods and co-supervised the experiments. S.K.-M.R.R. supervised research and performed the statistical analyses. İ.E. and O.G. conducted cultivation of the mono-culture and consortium, qPCR, FISH experiments and GC measurement. M.S. contributed medium design, mono-culture and FISH experiments. S.V. contributed to pre-culture and PCR experiments. B.H., G.B. and W.F. provided the HPLC method optimization and HPLC data. All authors provided critical feedback during the progress of the projects and helped to shape the research, analysis of data and helped to draft the initial version of the manuscript. İ.E. and S.K.-M.R.R. wrote and edited the final version of the manuscript.

## Competing interests

The authors declare no competing interests.
