## [Peer Review File · Communications Biology]

Reviewers' comments:

Reviewer #1 (Remarks to the Author):

The authors of the manuscript show that hydrogen evolution from glucose performed by a consortium of *Clostridium acetobutylicum* and *Enterobacter aerogenes* can be 40 % higher than the "Thauer limit", which was postulated by Thauer et al., 1977. The authors provide a clearly structured manuscript together with the notion that precision in e.g. media composition and a defined cell count of each bacterial culture lead to successful increase of hydrogen production in the described consortium. Since the extraordinary yield of hydrogen per mol of glucose has not been described before, this study would have a big impact on science and especially on biotechnological approaches. Therefore, this manuscript needs to be evaluated very thoroughly. Hence, next to some comprehensive questions, I like to address some major issues which need to be explained by the authors and which I would suggest also need to be changed or added to the manuscript.

Minor comments:

L. 38-57. The authors describe the importance of bacterial consortia, as well as the optimization of product yields for e.g. biotechnological approaches. This part is written very superficially as there are many generic sentences used but no examples given. To get a better overview, instead of only adding references, examples would help.

L. 42. ... multitude of possible metabolic reactions and interaction partners. (Which reactions and what kind of interactions?)

L. 44. Unrefined substrate (Which substrates?)

L. 44. ... environmental stressors (What kind of stressors?)

L. 49-57. A few references and examples regarding process optimization would be good.

L. 78. Thauer limit

L. 79-81. Please explain the PFL or PFOR pathways in more detail. Which reactions do these enzymes catalyze? Reaction equations would be helpful. Hydrogen is not the direct product of these reactions. Please specify the roles of e.g. formate hydrogen lyases and hydrogenases in hydrogen production. Jay et al., 2020. Integrated thermodynamic analysis of electron bifurcating [FeFe]-hydrogenase to inform anaerobic metabolism and H₂ production. *BBA Bioenergetics* 1861: 148087. (<https://doi.org/10.1016/j.bbabi.2019.148087>) gives a good overview of the most important findings for hydrogen generation in anaerobes, while taking the Thauer limit in account.

L. 82. The PFOR is operative in Clostridiaceae when regarding hydrogen formation

L. 198. Reischl et al., 2018- Reference has not been added to the reference list

L. 215. ... gas composition of serum bottles was determined.

L. 262. Supplementary figure 1: The OD600 needs to be displayed on a logarithmic scale (as for all graphs in which an OD600 is displayed). There are no error bars in figure d. Generally: How often were the growth experiments performed?

L.263. Please clarify the term "cumulative pressure". It has not been mentioned in the materials and methods part.

L. 276. Do I understand it right that the DoE settings A-I (except E) were done in triplicates and E was done in a pentaplicate?

L. 275-278. Since the media composition is very important for the reader, I would suggest to add Supplementary table 1 and Supplementary fig. 3 to the main manuscript.

L. 278-281. Due to the small size and choice of color scheme for Supplementary figure 3 it is hard to follow the author's conclusions on the growth experiments. I would suggest changing the colors scheme or using symbols instead.

L. 284. Please clarify: The data of which time point (or growth phase) was used in the model?

L. 289. In line 280 it is stated that media compositions C, H and I showed highest cumulative pressure. Medium I contains 150 mmol L⁻¹ PB. Please justify why results of the model calculations

indicate that lower buffer capacities relate to higher gas production.

L. 294. Reference number 47 does not link to a publication which explains the solventogenic switch in *C. acetobutylicum*.

L. 302. Fig. 1b

L. 312-313. The values of the concentrations for AC, SA and PB are mixed up.

L. 323-324. The letters a,b and c,d are missing in Fig. 2.

L. 536 All references need to be written in the same style, however, in some references, all authors are stated, in others, only one author and et al. is mentioned. Many species are not written in italics. The journal Appl. Environ. Microbiol. is abbreviated, all other journals are not.

L. 715. Delete Supplementary Fig. 4.

L. 716. Please state in the figure legend which consortium has been analyzed (1:10,000?). Fig. 2: The symbol for the HER is very big, hence the correct value is hard to identify. Please consider changing the symbol to an x or +. How often has the experiment been performed?

L. 740. Please state in the figure legend which consortium has been analyzed (1:10,000?).

L. 747. The legends of Table 1 and Table 2 need to be added. Please mention in the legend of Table 1 why these time points for calculation of the respective rates were chosen.

L. 758. Suppl. Table 7. Please clarify why the equations include the word "error". If there are unknown products, please state "error" as unknown product.

Major issues:

L. 194-196. ...remaining gas was released from the serum bottles. How was it released? How much pressure was left inside the bottle? Has the overpressure that was generated due to hydrogen and CO₂ production been measured?

L. 214-215. Was the pressure generated during gas production included for the HER calculations? This has not been clarified in the manuscript. The pressure needs to be taken in account for calculation of the hydrogen concentration in the headspace of a serum bottle.

L. 373. Please clarify whether the overpressure was released from the serum bottle after each gas measurement. This could also be considered as "active gas removal".

L. 369-372. After examining data shown in table 1 and table 2, it is unclear how those high hydrogen yields per glucose were calculated. This is important for the reader because these results lead to the major conclusion of the study. Table 2 displays the HER measured after certain time intervals and table 1 shows the glucose/cellobiose uptake rate for the same time point. For instance, in table 2 the highest Y(H₂/S) is 0.73 for growth on cellobiose, and I assume calculated with the HER of 7.48 mmol L⁻¹, which was sampled at time point 42.5 h. At the same time point the cellobiose uptake rate was 14.18 C mmol L⁻¹. Calculating $7.48 / 14.18 = 0.505$, I get a lower hydrogen yield, namely 3.03 mol(H₂) mol⁻¹(C₆ sugar equiv.).

When calculating the Y(H₂/S) during growth on glucose, table 2 shows highest Y(H₂/S) is 0.93 at time point (or interval) 53 h, which was, I assume, calculated with the HER of 4.04 mmol L⁻¹. At the same time point the glucose uptake rate (table 1) was 17.25 C-mmol L⁻¹. Calculating $4.04 / 17.25 = 0.234$, I get a lower hydrogen yield, namely 1.4 mol(H₂) mol⁻¹(glucose). I am also wondering why a very high HER of 10.31 mmol L⁻¹ was detected after 39.5 h but this was not considered for the Y(H₂/S) calculations.

Please clarify these major finding by adding calculation of the Y(H₂/S) to the main text. It would also be important for the reader of the article to be able to reconstruct the Y(H₂/S) calculations by displaying the glucose/cellobiose uptake rates together with the according HERs in the same table.

L. 451-454. In this paragraph the authors argue that cells of *C. acetobutylicum* have started to sporulate at time point 53 h due to a pH decrease during growth. Is there a figure displaying the changes in pH during growth? Please explain how sporulating *C. acetobutylicum* cells are able to produce (in a consortium with *E. aerogenes*) the highest ever known Y(H₂/S)?

Reviewer #2 (Remarks to the Author):

In the manuscript the authors reported an engineered microbial consortium system composed by *E. aerogenes* and *C. acetobutylicum*, that exhibited the highest reported H₂ yield per carbon substrate consumed (Y_{H2/S}), either in medium supplied with glucose or cellobiose. The calculated Y_{H2/S} was raised significantly than any other reports, higher than the Thauer limit, the result is encouraging. However, since a high Y_{H2/S} exceeds the Thauer limit has also been demonstrated previously, the study can be more scientifically significance if the authors could reveal the underlying mechanisms, or present a possible model(s) for what they have discovered using the specific consortium. Secondly, concerning the calculation of Y_{H2/S}, more concrete data or detailed calculation process shall be presented in the manuscript, since that is related to the novelty of the work.

Reply to reviewer comments

Reviewer #1

The authors of the manuscript show that hydrogen evolution from glucose performed by a consortium of *Clostridium acetobutylicum* and *Enterobacter aerogenes* can be 40 % higher than the “Thauer limit”, which was postulated by Thauer et al., 1977. The authors provide a clearly structured manuscript together with the notion that precision in e.g. media composition and a defined cell count of each bacterial culture lead to successful increase of hydrogen production in the described consortium. Since the extraordinary yield of hydrogen per mol of glucose has not been described before, this study would have a big impact on science and especially on biotechnological approaches. Therefore, this manuscript needs to be evaluated very thoroughly. Hence, next to some comprehensive questions, I like to address some major issues which need to be explained by the authors and which I would suggest also need to be changed or added to the manuscript.

Reply: *Thank you very much for your comments and suggestions. We took especially care that all the calculations and the material (and methods) become clear when reading the manuscript. We hope you find the improved version of our manuscript suitable for publications.*

Minor comments:

L. 38-57. The authors describe the importance of bacterial consortia, as well as the optimization of product yields for e.g. biotechnological approaches. This part is written very superficially as there are many generic sentences used but no examples given. To get a better overview, instead of only adding references, examples would help.

Reply: *Thank you for this comment. We think the introduction has certain flow and adding specific examples will disturb it. However, for your following comments we did some additions. We change the specified part as follows:*

“Microbial communities exist in high levels of biodiversity, enabling cooperation and interaction among its members in functional metabolic networks³. Compared to mono-cultures, a microbial consortium empowers complex metabolic tasks due to the multitude of possible metabolic reactions and interaction possibilities, which are based on mutualism, commensalism or neutralism^{4,5}. The streamlined syntrophic interactions or commensal relationships among the microorganisms in microbial consortia were shown to enable an efficient utilisation of unrefined substrates, such as cane molasses or beet molasses^{6,7}, to resist to environmental stressors, e.g. temperature fluctuations or heavy metal exposure⁷⁻⁹ and to display high productivity or yield^{10,11}. In nature, a modest undefined consortium may contain thousands of species¹². However, for efficiently performing bioconversions in natural or artificial ecosystems, the specific metabolic reactions of individual species in the consortium are more relevant than the species richness¹³”

L. 42. ... multitude of possible metabolic reactions and interaction partners. (Which reactions and what kind of interactions?)

Reply: *Thank you for your comment. We adapted the sentence as “Compared to mono-cultures, a microbial consortium empowers complex metabolic tasks due to the multitude of possible metabolic reactions and interaction possibilities, which are based on mutualism, commensalism or neutralism.”*

L. 44. Unrefined substrate (Which substrates?)

Reply: Thank you for your comment. We added examples. The sentence was changed as “The streamlined syntrophic interactions or commensal relationships among the microorganisms in microbial consortia were shown to enable an efficient utilisation of unrefined substrates, such as cane molasses or beet molasses^{6,7}, to resist to environmental stressors, e.g. temperature fluctuations or heavy metal exposure⁷⁻⁹ and to display high productivity or yield^{10,11}”

L. 44. ... environmental stressors (What kind of stressors?)

Reply: Thank you for your comment. We added examples “to resist to environmental stressors, e.g. temperature fluctuations or heavy metal exposure”.

L. 49-57. A few references and examples regarding process optimization would be good.

Reply: Thank you for this comment. This was meant as the introduction paragraph and the specific examples and references can be found in the next paragraph. I hope that suits you.

L. 78. Thauer limit

Reply: Thank you for your comment. We changed the sentence as; “According to the theoretical limit, the so called “Thauer limit”, 4 moles H₂ can be produced per one mole of glucose consumed during dark fermentation when acetate is produced as by-product”

L. 79-81. Please explain the PFL or PFOR pathways in more detail. Which reactions do these enzymes catalyze? Reaction equations would be helpful. Hydrogen is not the direct product of these reactions. Please specify the roles of e.g. formate hydrogen lyases and hydrogenases in hydrogen production. Jay et al., 2020. Integrated thermodynamic analysis of electron bifurcating [FeFe]-hydrogenase to inform anaerobic metabolism and H₂ production. BBA Bioenergetics 1861: 148087. (<https://doi.org/10.1016/j.bbabi.2019.148087>) gives a good overview of the most important findings for hydrogen generation in anaerobes, while taking the Thauer limit in account.

Reply: Thank you for your comment. Detailed information was added in the manuscript as follows “Depending on the microbial group, H₂ formation may occur either via the pyruvate-formate-lyase (PFL) pathway or the pyruvate ferredoxin oxidoreductase (PFOR) pathway³². The PFL pathway is operative in Enterobacteriaceae. In this pathway, pyruvate is converted into acetyl-CoA and formate. Formate is either shuttled out of the cell or it can be split into carbon dioxide (CO₂) and H₂ by formate hydrogen lyase³². The PFOR pathway is operative in Clostridiaceae and H₂ production occurs through the action of [NiFe]- and/or [FeFe]-hydrogenases^{36,37}.”.

L. 82. The PFOR is operative in Clostridiaceae when regarding hydrogen formation

Reply: Thank you for your comment. We added “during H₂ production” in to the sentence.

L. 198. Reischl et al., 2018- Reference has not been added to the reference list

Reply: Thank you for your comment. We added that to the references.

L. 215. ... gas composition of serum bottles was determined.

Reply: Thank you for your comment. The sentence was changed to “The $Y(H_2/S)$ [mol mol⁻¹], HER [mmol L⁻¹ h⁻¹] and the specific H_2 production rate (q_{H_2}) [mmol g⁻¹ h⁻¹]³² were calculated from the intervals between each time-point and the gas composition in the headspace of serum bottle was determined using the GC ”.

L. 262. Supplementary figure 1: The OD600 needs to be displayed on a logarithmic scale (as for all graphs in which an OD600 is displayed). There are no error bars in figure d. Generally: How often were the growth experiments performed?

Reply: Thank you for your comment. We improved the graph. Please find it in the supplementary information file. Other absorbance graphs were also changed. Experiments were replicated three times (three different sets), hence $N = 3$ and each set contained quadruplicate ($n = 4$).

L.263. Please clarify the term “cumulative pressure”. It has not been mentioned in the materials and methods part.

Reply: Thank you for your comment. We added the explanation in methods – closed batch cultivation section- as follows: “. The pressure in the headspace of the serum bottles were measured individually using a manometer (digital manometer LEO1-Ei, –1...3 bar, Keller, Germany). After each measurement, the pressure was released completely from the headspace of serum bottle by penetrating the butyl rubber stopper with a sterile needle. The pressure values were added up to reveal total produced pressure (cumulative pressure)”.

L. 276. Do I understand it right that the DoE settings A-I (except E) were done in triplicates and E was done in a pentaplicate?

Reply: Thank you for your comment. Yes, middle point which is set E was done pentaplicate ($n = 5$)

L. 275-278. Since the media composition is very important for the reader, I would suggest to add Supplementary table 1 and Supplementary fig. 3 to the main manuscript.

Reply: Thank you for your comment. We added the Supplementary Table 1 as Table 1 and Supplementary Fig 3 as Figure 1 to the main text.

L. 278-281. Due to the small size and choice of color scheme for Supplementary figure 3 it is hard to follow the author’s conclusions on the growth experiments. I would suggest changing the colors scheme or using symbols instead.

Reply: Thank you for your comment. Supplementary figure 3 was changed according to suggestions and added to main text as Fig. 1.

L. 284. Please clarify: The data of which time point (or growth phase) was used in the model?

Reply: Thank you for your comment. Final time point (end of exponential phase) was used for model generation. We added this information on the Design of Experiment section to the manuscript.

L. 289. In line 280 it is stated that media compositions C, H and I showed highest cumulative

pressure. Medium I contains 150 mmol L-1 PB. Please justify why results of the model calculations indicate that lower buffer capacities relate to higher gas production.

Reply: *Thank you for your comment. It is somehow difficult to justify an outcome of a statistical test. Either the test is significant, as in our case, or not. Hence the lower PB capacity has a significant effect on H₂ production.*

L. 294. Reference number 47 does not link to a publication which explains the solventogenic switch in *C. acetobutylicum*.

Reply: *Thank you for your comment. We changed the reference.*

L. 302. Fig. 1b

Reply: *Thank you for your comment. We changed "Fig. 1" to "Fig. 1b".*

L. 312-313. The values of the concentrations for AC, SA and PB are mixed up.

Reply: *Thank you for your comment. We corrected the concentrations.*

L. 323-324. The letters a,b and c,d are missing in Fig. 2.

Reply: *Thank you for your comment. Corrected.*

L. 536 All references need to be written in the same style, however, in some references, all authors are stated, in others, only one author and et al. is mentioned. Many species are not written in italics. The journal Appl. Environ. Microbiol. is abbreviated, all other journals are not.

Reply: *Thank you for your comment. We corrected the citation style throughout the manuscript.*

L. 715. Delete Supplementary Fig. 4.

Reply: *Thank you for your comment. Supplementary Fig 4 was deleted.*

L. 716. Please state in the figure legend which consortium has been analyzed (1:10,000?). Fig. 2: The symbol for the HER is very big, hence the correct value is hard to identify. Please consider changing the symbol to an x or +. How often has the experiment been performed?

Reply: *Thank you for your comment. The symbol was changed. We added "(with an inoculum ratio of 1:10,000 *E. aerogenes*:*C. acetobutylicum*)" in to the legend. Experiment was performed 3 times.*

L. 740. Please state in the figure legend which consortium has been analyzed (1:10,000?).

Reply: *Thank you for your comment. . We added "(with an inoculum ratio of 1:10,000 *E. aerogenes* : *C. acetobutylicum*)" in to the legend.*

L. 747. The legends of Table 1 and Table 2 need to be added. Please mention in the legend of Table 1 why these time points for calculation of the respective rates were chosen.

Reply: Thank you for your comment. We added the explanation of the term “global” into the methods – data analysis – section. We also added a footnote to Table 2 (old Table 1) as “Substrate uptake rate, by-product production rates and the mass balance analyses were conducted between first and last time points (Global).” and a footnote to Table 3 (old Table 2) as “Values were calculated in between time points (after gas production started) as indicated in the table.” We also changed the time points in the table.

L. 758. Suppl. Table 7. Please clarify why the equations include the word “error”. If there are unknown products, please state “error” as unknown product.

Reply: Thank you for your comment. We changed “error” to “unknown product”

Major issues:

L. 194-196. ...remaining gas was released from the serum bottles. How was it released? How much pressure was left inside the bottle? Has the overpressure that was generated due to hydrogen and CO₂ production been measured?

Reply: Thank you for your comment. Explanation was added in the manuscript as follows; “After every GC measurement, remaining gas was released completely from the serum bottles by sticking the sterile needle in to the blue stoppers. The serum bottles were checked/measured whether if there is any remaining gas at the head space of the serum bottles by a manometer (digital manometer LEO1-Ei, -1...3 bar, Keller, Germany).”, and the serum bottles were put back to incubator to continue the experiments. Until the next time point, they stayed in the incubator and same process was followed: pressure was measured, GC measurements were done, remaining gas was released, and bottles were put back to incubator.

L. 214-215. Was the pressure generated during gas production included for the HER calculations? This has not been clarified in the manuscript. The pressure needs to be taken in account for calculation of the hydrogen concentration in the headspace of a serum bottle.

Reply: Thank you for your comment. Yes, as you already pointed out, it would not have been possible to measure the H₂ concentration in the head space of the serum bottle otherwise. We added the measurement of pressure in methods.

L. 373. Please clarify whether the overpressure was released from the serum bottle after each gas measurement. This could also be considered as “active gas removal”.

Reply: Thank you for your comment. To clarify, we changed “active gas removal techniques” with “automated gas removal techniques”. We released the gas manually; this was the reason we pointed out without “active gas removal”. However, for clarification the wording was changed.

L. 369-372. After examining data shown in table 1 and table 2, it is unclear how those high hydrogen yields per glucose were calculated. This is important for the reader because these results lead to the major conclusion of the study. Table 2 displays the HER measured after certain time intervals and table 1 shows the glucose/cellobiose uptake rate for the same time point. For instance, in table 2 the highest Y(H₂/S) is 0.73 for growth on cellobiose, and I assume calculated with the HER of 7.48 mmol L⁻¹, which was sampled at time point 42.5 h. At the same time point the cellobiose uptake rate was

14.18 C mmol L⁻¹. Calculating $7.48 / 14.18 = 0.505$, I get a lower hydrogen yield, namely 3.03 mol(H₂) mol⁻¹(C₆ sugar equiv.).

When calculating the Y(H₂/S) during growth on glucose, table 2 shows highest Y(H₂/S) is 0.93 at time point (or interval) 53 h, which was, I assume, calculated with the HER of 4.04 mmol L⁻¹. At the same time point the glucose uptake rate (table 1) was 17.25 C-mmol L⁻¹. Calculating $4.04 / 17.25 = 0.234$, I get a lower hydrogen yield, namely 1.4 mol(H₂) mol⁻¹(glucose). I am also wondering why a very high HER of 10.31 mmol L⁻¹ was detected after 39.5 h but this was not considered for the Y(H₂/S) calculations.

Please clarify these major finding by adding calculation of the Y(H₂/S) to the main text. It would also be important for the reader of the article to be able to reconstruct the Y(H₂/S) calculations by displaying the glucose/cellobiose uptake rates together with the according HERs in the same table.

Reply: Thank you for your comment. Substrate uptake rates and by-product production rates were calculated globally. Calculation of global substrate uptake rate, by-product production rates and the mass balance analyses of the mono-cultures and consortium on glucose and cellobiose were done between first and last time point. This explanation is now indicated in Data analysis section in methods and shown in Table 2. However, the HER values and the Y(H₂/S) values were calculated from the intervals. Hence, HER of 4.04 shown in Table 3, is actually HER value calculated in between time point 39.5 and 53, not from time point 0 to time point 53. However, the value of 17.25 C-mmol L⁻¹ glucose uptake rate (Table 2) was calculated between time-point 0 and 53. So dividing 4.04 by 17.25 would not represent the Y(H₂/S) for that particular time-point.

In that sense, calculating Y(H₂/S) should be HER/substrate uptake rate in that particular time point (between 39.5 and 53 h). At this point; We unfortunately cannot **only** consider the glucose (or cellobiose) as only substrate uptake in that particular time point. As it can be seen in Supplementary Table 6, left side of the equation contains not only glucose (or cellobiose). Because other by-products (e.g. formic acid or ethanol) were produced (e.g. until time point 39.5), but thereafter, they were consumed (e.g. from time point 39.5 to 53). In that sense, the particular time point example you pointed out (time point 53 which represents 13.5 hours – between time points 39.5-53), we have to consider by-product re-uptake as well. Ultimately, it is a defined medium, meaning glucose (or cellobiose) is only substrate but in between time points we cannot only consider glucose uptake rate between time point 39.5 and time point 53. So the total consumed compounds were 4.34 (glucose + consumed by-products between 39.5-53 h). When we divide 4.04 by 4.34; we obtain a yield of “0.93” produced H₂ between “39.5-53 h”. The reaction details can be found in Supplementary Table 6.

We also added the between time-points to both Table 2 and 3.

L. 451-454. In this paragraph the authors argue that cells of *C. acetobutylicum* have started to sporulate at time point 53 h due to a pH decrease during growth. Is there a figure displaying the changes in pH during growth? Please explain how sporulating *C. acetobutylicum* cells are able to produce (in a consortium with *E. aerogenes*) the highest ever known Y(H₂/S)?

Reply: Thank you for your comment. We added pH values in Supplementary Figure 6. After 53 h; there was no gas production and sporulation occurred after 53h. Also decrease in the number of active cells of *C. acetobutylicum* can be observed in the same figure (T4 (39.5h) and T5 (53h)). Initial experiments of FISH showed that probes cannot bind to cells of *C. acetobutylicum* if they formed spores (data not shown). However, we did not perform particular experiments to investigate sporulation.

Reviewer #2

In the manuscript the authors reported an engineered microbial consortium system composed by *E. aerogenes* and *C. acetobutylicum*, that exhibited the highest reported H₂ yield per carbon substrate consumed (Y_{H₂/S}), either in medium supplied with glucose or cellobiose. The calculated Y_{H₂/S} was raised significantly than any other reports, higher than the Thauer limit, the result is encouraging. However, since a high Y_{H₂/S} exceeds the Thauer limit has also been demonstrated previously, the study can be more scientifically significance if the authors could reveal the underlying mechanisms, or present a possible model(s) for what they have discovered using the specific consortium. Secondly, concerning the calculation of Y_{H₂/S}, more concrete data or detailed calculation process shall be presented in the manuscript, since that is related to the novelty of the work.

Reply: Thank you very much for your comments. We improved the manuscript according to your comments by adding additional information and explaining the details of calculations in the methods section. Please find two explanations to your questions below. We hope you find the explanations satisfactory and hope that the improved version of our manuscript will make the experimental approach and the calculations clear to any reader.

To your first comment; The demonstrated “high Y_{H₂/S} exceeds the Thauer limit” was in a complex medium (containing other carbon sources such as yeast extract). Here we represent a defined medium and only carbon source for respected microorganisms is either glucose or cellobiose. Supplementary Table 6 revealing not only glucose uptake but also by-products (which are produced during the fermentation) uptake rates. Consortium converted previously produced by products to H₂. Our encouraging results are also reflected in our C-balance and Degree of reduction balance calculations, which support the outcome on a high Y_(H₂/S) as well as the supplementary table 6 which contains the reaction details and all equations.

Concerning your second comment; we added the explanation of how we calculated the Y(H₂/S) and HER as follows; “The Y(H₂/S) [mol mol⁻¹], HER [mmol L⁻¹ h⁻¹] and the specific H₂ production rate (qH₂) [mmol g⁻¹ h⁻¹] were calculated from the intervals between each time-point and the gas composition in the headspace of serum bottle was determined using the GC.”, as well as the calculation of “global” and “between time-points” in text and in the Table 2 and 3.

REVIEWERS' COMMENTS:

Reviewer #1 (Remarks to the Author):

The authors provide in their resubmission of the manuscript 'Biohydrogen production beyond the Thauer limit by precision design of artificial microbial consortia' sufficient information and corrected the manuscript thoroughly according to my major and minor comments. Therefore, I am now willing to accept the corrected manuscript for publication in the journal Communications Biology.